# Self-Supervised Generation of Spatial Audio for 360° Video

**Pedro Morgado**
University of California, San Diego*

**Nuno Vasconcelos**
University of California, San Diego

**Timothy Langlois**
Adobe Research, Seattle

**Oliver Wang**
Adobe Research, Seattle

## Abstract

We introduce an approach to convert mono audio recorded by a 360° video camera into *spatial audio*, a representation of the distribution of sound over the full viewing sphere. Spatial audio is an important component of immersive 360° video viewing, but spatial audio microphones are still rare in current 360° video production. Our system consists of end-to-end trainable neural networks that separate individual sound sources and localize them on the viewing sphere, conditioned on multi-modal analysis of audio and 360° video frames. We introduce several datasets, including one filmed ourselves, and one collected in-the-wild from YouTube, consisting of 360° videos uploaded *with* spatial audio. During training, ground-truth spatial audio serves as self-supervision and a mixed down mono track forms the input to our network. Using our approach, we show that it is possible to infer the spatial location of sound sources based only on 360° video and a mono audio track.

## 1 Introduction

360° video provides viewers an immersive viewing experience where they are free to look in any direction, either by turning their heads with a Head-Mounted Display (HMD), or by mouse-control while watching the video in a browser (e.g., YouTube). Capturing 360° video involves filming the scene with multiple cameras and stitching the result together. While early systems relied on expensive rigs with carefully mounted cameras, recent consumer-level devices combine multiple lenses in a small fixed-body frame that enables automatic stitching, allowing 360° video to be recorded with a single push of a button.

As humans rely on audio localization cues for full scene awareness, *spatial audio* is a crucial component of 360° video. Spatial audio enables viewers to experience sound in all directions, while adjusting the audio in real time to match the viewing position. This gives users a more immersive experience, as well as providing cues about which part of the scene might have interesting content to look at. However, unlike 360° video, producing spatial audio content still requires a moderate degree of expertise. Most consumer-level 360° cameras only record mono audio, and syncing an external spatial audio microphone can be expensive and technically challenging. As a consequence, while most video platforms (e.g., YouTube and Facebook) support spatial audio, it is often ignored by content creators, and at the time of submission, a random polling of 1000 YouTube 360° videos yielded *less than 5%* with spatial audio.

In order to close this gap between the audio and visual experiences, we introduce three main contributions: (1) we formalize the *360° spatialization* problem; (2) design the first 360° spatialization procedure; and (3) collect two datasets and propose an evaluation protocol to benchmark ours and

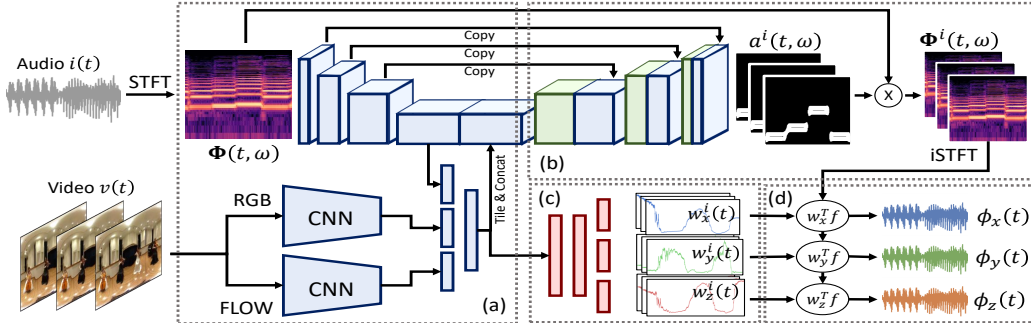

**Figure 1: Architecture overview**. Our approach is composed of four main blocks. The input video and audio signals are fed into the analysis block (a), which extracts high-level features. The separation block (b) then learns $k$ time-frequency attenuation maps $a^i(t, w)$ to modulate the input STFT and produce modified waveforms $f^i(t)$. The localization block (c) computes a set of linear transform weights $w^i(t)$ that localize each source. In the ambisonics generation step (d), localization weights are then combined with the separated sound sources to produce the final spatial audio output.

future algorithms. $360°$ spatialization aims to *upconvert a single mono recording into spatial audio guided by full 360 view video*. More specifically, we seek to generate spatial audio in the form of a popular encoding format called first-order ambisonics (FOA), given the mono audio and corresponding $360°$ video as inputs. In addition to formulating the $360°$ spatialization task, we design the first data-driven system to upgrade mono audio using self-supervision from $360°$ videos recorded with spatial audio. The proposed procedure is based on a novel neural network architecture that disentangles two fundamental challenges in audio spatialization: the separation of sound sources from a mixed audio input and respective localization of these sources. In order to train and validate our approach, we introduce two $360°$ video datasets with spatial audio, one recorded by ourselves in a constrained domain, and a large-scale dataset collected *in-the-wild* from YouTube. During training, the captured spatial audio serves as ground truth, with a mixed down mono version provided as input to our system. Experiments conducted in both datasets show that the proposed neural network can generate plausible spatial audio for $360°$ video. We further validate each component of the proposed architecture and show its superiority over a state-of-the-art, but domain-independent baseline architecture.

In the interest of reproducibility, code, data and trained models will be made available to the community at https://pedro-morgado.github.io/spatialaudiogen.

## 2   Related Work

To the best of our knowledge, we propose the first system for audio spatialization. In addition to spatial audio, the fields most related to our work are self-supervised learning, audio generation, source separation and audio-visual cross-modal learning, which we now briefly describe.

**Spatial audio**   Artificial environments, such as those rendered by game engines, can play sounds from any location in the video. This capability requires recording sound sources separately and mixing them according to the desired scene configuration (i.e., the positions of each source relative to the user). In a real world recording, however, sound sources cannot be recorded separately. In this case where sound sources are naturally mixed, spatial audio is often encoded using Ambisonics [13, 9, 30].

Ambisonics aim to approximate the sound pressure field at a single point in space using a spherical harmonic decomposition. More specifically, an audio signal $f(\boldsymbol{\theta}, t)$ arriving from direction $\boldsymbol{\theta} = (\varphi, \vartheta)$ (where $\varphi$ is the zenith angle and $\vartheta$ the azimuth angle) at time $t$ is represented by a truncated spherical harmonic expansion of order $N$

$$f(\boldsymbol{\theta}, t) = \sum_{n=0}^{N} \sum_{m=-n}^{n} Y_n^m(\varphi, \vartheta) \phi_n^m(t) \tag{1}$$

where $Y_n^m(\varphi, \vartheta)$ is the real spherical harmonic of order $n$ and degree $m$, and $\phi_n^m(t)$ are the coefficients of the expansion. For ease of notation, $Y_n^m$ and $\phi_n^m$ can be stacked into vectors $\boldsymbol{y}_N$ and $\boldsymbol{\phi}_N$, and (Eq. 1) written as $f(\boldsymbol{\theta}, t) = \boldsymbol{y}_N^T(\boldsymbol{\theta}) \boldsymbol{\phi}_N(t)$.

In a controlled environment, sound sources with *known* locations can be synthetically encoded into ambisonics using their spherical harmonic projection. More specifically, given a set of $k$ audio signals $s_1(t), \ldots, s_k(t)$ originating from directions $\boldsymbol{\theta}_1, \ldots, \boldsymbol{\theta}_k$,

$$\boldsymbol{\phi}_N(t) = \sum_{i=1}^{k} \boldsymbol{y}_N(\boldsymbol{\theta}_i) s_i(t). \tag{2}$$

For ambisonics playback, $\boldsymbol{\phi}_N$ is then *decoded* into a set of speakers or headphone signals in order to provide a plane-wave reconstruction of the sound field. In sum, the coefficients $\boldsymbol{\phi}_N$, also known as *ambisonic channels*, are sufficient to encode and reproduce spatial audio. Hence, our goal is to generate $\boldsymbol{\phi}_N$ from non-spatial audio and the corresponding video.

**Self-supervised learning**  Neural networks have been successfully trained through self-supervision for tasks such as image super-resolution [10, 27] and image colorization [20, 46]. In the audio domain, self-supervision has also enabled the detection of sound-video misalignment [37] and audio super-resolution [31]. Inspired by these approaches, we propose a self-supervised technique for audio spatialization. We show that the generation of ambisonic audio can be learned using a dataset of 360° video with spatial audio collected in-the-wild without additional human intervention.

**Generative models**  Recent advances in generative models such as Generative Adversarial Networks (GANs) [14] or Variational Auto-Encoders (VAE) [29] have enabled the generation of complex patterns, such as images [14] or text [23]. In the audio domain, Wavenet [36] has demonstrated the ability to produce high fidelity audio samples of both speech and music, by generating a waveform from scratch on a sample-by-sample basis. Furthermore, neural networks have also outperformed prior solutions to audio super-resolution [31] (e.g. converting from 4kHz to 16kHz audio) using a U-Net encoder-decoder architecture, and have enabled "automatic-Foley" type applications [41, 38], i.e. generating sounds that correspond to image features, and vice-versa. In this work, instead of generating audio from scratch, our goal is to augment the input audio channels so as to introduce spatial information. Thus, unlike Wavenet, efficient audio generation can be achieved without sacrificing audio fidelity, by transforming the input audio. We also demonstrate the advantages of our approach, inspired by the ambisonics encoding process in controlled environments, over a generic U-Net architecture for spatial audio generation.

**Source separation**  Source separation is a classic problem with an extensive literature. While early methods present the problem as independent component analysis, and focused on maximizing the statistical independence of the extracted signals [24, 7, 6, 2], recent approaches focus on data-driven solutions. For example, [19] proposes a recurrent neural-network for monaural separation of two speakers, [1, 12, 11] seek to isolate sound sources by leveraging synchronized visual information in addition to the audio input, and [44] studies a wide range of frequency-based separation methods. Similarly to recent trends, we rely on neural networks guided by cross-modal video analysis. However, instead of only separating human speakers [44] or musical instruments [47], we aim to separate multiple unidentified types of sound sources. Also, unlike previous algorithms, no explicit supervision is available to learn the separation block.

**Source localization**  Sound source localization is a mature area of signal processing and robotics research [3, 35, 34, 42]. However, unlike the proposed 360° spatialization problem, these works rely on microphone arrays using beamforming techniques [43] or binaural audio and HRTF cues similar to those used by humans [18]. Furthermore, the need for carefully calibrated microphones limits the applicability of these techniques to videos collected in-the-wild.

**Cross visual-audio analysis**  Cross-modal analysis has been extensively studied in the vision and graphics community, due to the inherently paired nature of video and audio. For example, [4] learns audio feature representations in an unsupervised setting by leveraging synchronized video. [22] segments and localizes dominant sound sources using clustering of video and sound features. Other methods correlate repeated motions with sounds to identify sound sources such as the strumming of a guitar using for example canonical correlation analysis [25, 26], joint embedding spaces [41, 38] or other temporal features [5].

# 3 Method

In this section, we define the 360° spatialization task to upconvert common audio recordings to support spatial audio playback. We then introduce a deep learning architecture to address this task, and two datasets to train the proposed architecture.

## 3.1 Audio spatialization

The goal of 360° spatialization is to generate ambisonic channels $\boldsymbol{\phi}_N(t)$ from non-spatial audio $i(t)$ and corresponding video $v(t)$. To handle the most common audio formats supported by commercial 360° cameras and video viewing platforms (e.g., YouTube and Facebook), we upgrade monaural recordings (mono) into first-order ambisonics (FOA). FOA consists of four channels that store the first-order coefficients, $\phi_0^0, \phi_1^{-1}, \phi_1^0$ and $\phi_1^1$, of the spherical harmonic expansion in (Eq. 1). For ease of notation, we refer to these tracks as $\phi_w, \phi_y, \phi_z$ and $\phi_x$, respectively.

**Self-supervised audio spatialization**    Converting mono to FOA ideally requires learning from videos with paired mono and ambisonics recordings, which are difficult to collect in-the-wild. In order to learn from self-supervision, we assume that monaural audio is recorded with an omni-directional microphone. Under this assumption, mono is equivalent to zeroth-order ambisonics (up to an amplitude scale) and, as a consequence, the upconversion only requires the synthesis of the missing higher-order channels. More specifically, we learn to predict the first-order components $\phi_x(t), \phi_y(t), \phi_z(t)$ from the (surrogate) mono audio $i(t) = \phi_w(t)$ and video input $v(t)$. Note that the proposed framework is also applicable to other conversion scenarios, e.g. FOA to second-order ambisonics (SOA), simply by changing the number of input and output audio tracks (see Sec 5).

## 3.2 Architecture

Audio spatialization requires solving two fundamental problems: source separation and localization. In controlled environments, where the separated sound sources $s_i(t)$ and respective localization $\boldsymbol{\theta}_i$ are known in advance, ambisonics can be generated using (Eq. 2). However, since $s_i(t)$ and $\boldsymbol{\theta}_i$ are not known in practice, we design dedicated modules to isolate sources from the mixed audio input and localize them in the video. Also, because audio and video are complementary for identifying each source, both separation and localization modules are guided by a multi-modal audio-visual analysis module. A schematic description of our architecture is shown in Fig. 1. We now describe each component. Details of network architectures are provided in Appendix A.

**Audio and visual analysis**    Audio features are extracted in the time-frequency domain, which has produced successful audio representations for tasks such as audio classification [17] and speaker identification [33]. More specifically, we extract a sequence of short-term Fourier transforms (STFT) computed on 25ms segments of the input audio with 25% hop size and multiplied by Hann window functions. Then, we apply a (two-dimensional) CNN encoder to the audio spectrogram, which progressively reduces the spectrogram dimensionality and extracts high-level features.

Video features are extracted using a two-stream network, based on Resnet-18 [16], to encode both appearance (RGB frames) and motion (optical flow predicted by FlowNet2 [21]). Both streams are initialized with weights pre-trained on ImageNet [8] for classification, and fine-tuned on our task.

A joint audio-visual representation is then obtained by merging the three feature maps (audio, RGB and flow) produced at each time $t$. Since audio features are extracted at a higher frame rate than video features, we first synchronize the audio and video feature maps by nearest neighbor up-sampling of video features. Each feature map is then projected into a feature vector (1024 for audio and 512 for RGB and flow), and the outputs concatenated and fed to the separation and localization modules.

**Audio separation**    Although the number of sources may vary, this is often small in practice. Furthermore, psycoaccoustic studies have shown that humans can only distinguish a small number of simultaneous sources (three according to [39]). We thus assume an upper-bound of $k$ simultaneous sources, and implement a separation network that extracts $k$ audio tracks $f^i(t)$ from the input audio $i(t)$. The separation module takes the form of a U-Net decoder that progressively restores the STFT dimensionality through a series of transposed convolutions and skip connections from the audio

analysis stage of equivalent resolution. Furthermore, to visually guide the separation module, we concatenate the multi-modal features to the lowest resolution layer of the audio encoder. In the last up-sampling layer, we produce $k$ sigmoid activated maps $a^i(t, \omega)$, which are used to modulate the STFT of the mono input $\boldsymbol{\Phi}(t; \omega)$. The STFT of the $i^{th}$ source $\boldsymbol{\Phi}^i(t; \omega)$ is thus obtained through the soft-attention mechanism $\boldsymbol{\Phi}^i(t; \omega) = a^i(t, \omega) \cdot \boldsymbol{\Phi}(t; \omega)$, and the separated audio track $f^i(t)$ reconstructed as the inverse STFT of $\boldsymbol{\Phi}^i(t; \omega)$ using an overlap-add method.

**Localization**    To localize the sounds $f^i(t)$ extracted by the separation network, we implement a module that generates, at each time $t$, the localization weights $\mathbf{w}^i(t) = (w_x^i(t), w_y^i(t), w_z^i(t))$ associated with each of the $k$ sources, through a series of fully-connected layers applied to the multi-modal feature vectors of the analysis stage. In a parallel to the encoding mechanism of (Eq. 2) used in controlled environments, $\mathbf{w}^i(t)$ can be interpreted as the spherical harmonics $\boldsymbol{y}_N(\boldsymbol{\theta}_i(t))$ evaluated at the predicted position of the $i^{th}$ source $\boldsymbol{\theta}_i(t)$.

**Ambisonic generation**    Given the localization weights $\mathbf{w}^i(t)$ and separated wave-forms $f^i(t)$, the first-order ambisonic channels $\boldsymbol{\phi}(t) = (\phi_x(t), \phi_y(t), \phi_z(t))$ are generated by $\boldsymbol{\phi}(t) = \sum_{i=1}^{k} \mathbf{w}^i(t) f^i(t)$. In summary, we split the generation task into two components: generating the attenuation maps $a^i(t, \omega)$ for source separation, and the localization weights $\mathbf{w}^i(t)$. As audio is not generated from scratch, but through a transformation of the original input inspired by the encoding framework of (Eq. 2), we are able to achieve fast deployment speeds with high quality results.

### 3.3   Evaluation metrics

Let $\boldsymbol{\phi}(t)$ and $\hat{\boldsymbol{\phi}}(t)$ be the ground-truth and predicted ambisonics, and $\boldsymbol{\Phi}(t; \omega)$ and $\hat{\boldsymbol{\Phi}}(t; \omega)$ their respective STFTs. We now discuss several metrics used for evaluating the generated signals $\hat{\boldsymbol{\phi}}(t)$.

**STFT distance**    Our network is trained end-to-end to minimize errors between STFTs, i.e.,

$$MSE_{\text{stft}} = \sum_{p \in \{x,y,z\}} \sum_t \sum_\omega \| \Phi_p(t, \omega) - \hat{\Phi}_p(t, \omega) \|^2, \tag{3}$$

where $\| \cdot \|$ is the euclidean complex norm. $MSE_{\text{stft}}$ has well-defined and smooth partial derivatives and, thus, it is a suitable loss function. Furthermore, unlike the euclidean distance between raw waveforms, the STFT loss is able to separate the signal into its frequency components, which enables the network to learn the easier parts of the spectrum without distraction from other errors.

**Log-spectral distance (LSD)**    Distances that only compare the smoothed spectral behavior of audio signals are widely used throughout the audio literature. We use the log-spectral distance [15] between $\boldsymbol{\Phi}(t; \omega)$ and $\hat{\boldsymbol{\Phi}}(t; \omega)$, which measures the distance in dB between the two spectrograms using

$$LSD = \sum_{p \in \{x,y,z\}} \sum_t \sqrt{\frac{1}{K} \sum_{\omega=1}^{K} \left( 10 \log_{10} \left| \frac{\Phi_p(t, \omega)}{\hat{\Phi}_p(t, \omega)} \right| \right)^2}. \tag{4}$$

**Envelope distance (ENV)**    Due to the high-frequency nature of audio and the human insensitivity to phase differences, frame-by-frame comparison of raw waveforms do not capture perceptual similarity of two audio signals. Instead, we measure the euclidean distance between *envelopes* of $\boldsymbol{\phi}(t)$ and $\hat{\boldsymbol{\phi}}(t)$, where the envelope of an audio wave is computed using the Hilbert transform method [40].

**Earth Mover's Distance (EMD)**    Ambisonics model the sound field $f(\boldsymbol{\theta}, t)$ over all directions $\boldsymbol{\theta}$. The energy of the sound field measured over a small window $w_t$ around time $t$ along direction $\boldsymbol{\theta}$ is

$$E(\boldsymbol{\theta}, t) = \sqrt{\frac{1}{T} \sum_{\tau \in w_t} f(\boldsymbol{\theta}, \tau)^2} = \sqrt{\frac{1}{T} \sum_{\tau \in w_t} \left( \boldsymbol{y}_N^T(\boldsymbol{\theta}) \boldsymbol{\phi}_N(\tau) \right)^2}. \tag{5}$$

Thus, $E(\boldsymbol{\theta}, t)$ represents the directional energy map of $\boldsymbol{\phi}(t)$. In order to measure the *localization* accuracy of the generated spatial audio, we propose to compute the EMD [32] between the energy maps $E(\boldsymbol{\theta}, t)$ associated with $\boldsymbol{\phi}(t)$ and $\hat{\boldsymbol{\phi}}(t)$. In practice, we uniformly sample the maps $E(\boldsymbol{\theta}, t)$ over the sphere, normalize the sampled map so that $\sum_i E(\boldsymbol{\theta}_i, t) = 1$, and measure the distance between samples over the sphere's surface using cosine (angular) distances for EMD calculation.

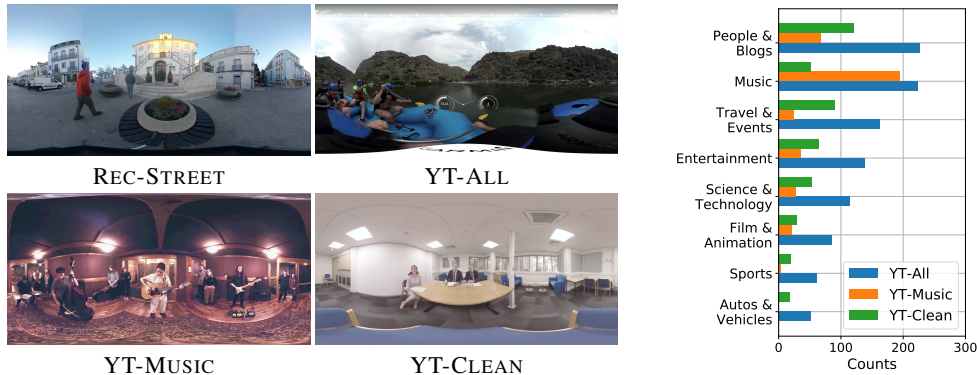

RE C-STREET                    YT-ALL

YT-MUSIC                       YT-CLEAN

**Figure 2: Representative images.** Example video frames from each dataset.

## 3.4 Datasets

To train our model, we collected two datasets of $360°$ videos with FOA audio. The first dataset, denoted REC-STREET, was recorded by us using a Theta V $360°$ camera with an attached TA-1 spatial audio microphone. REC-STREET consists of 43 videos of outdoor street scenes, totaling 3.5 hours and 123k training samples (0.1s each). Due to the consistency of capture hardware and scene content, the audio of REC-STREET videos is relatively easier to spatialize.

The second dataset, denoted YT-ALL, was collected in-the-wild by scraping $360°$ videos from YouTube using queries related to spatial audio, e.g., *spatial audio, ambisonics*, and *ambix*. To clean the search results, we automatically removed videos that did not contain valid ambisonics, as described by YouTube's format, keeping only videos containing all 4 channels or with only the Z channel missing (a common spatial audio capture scenario). Finally, we performed a manual curation to remove videos containing 1) still images, 2) computer generated content, or 3) post-processed and non-visually indicated sounds such as background music or voice-overs. During this pruning process, 799 videos were removed, resulting in 1146 valid videos totaling 113.1 hours of content (3976k training samples). YT-ALL was further separated into live musical performances, YT-MUSIC (397 videos), and videos with a small number of super-imposed sources which could be localized in the image, YT-CLEAN (496 videos). Upgrading YT-MUSIC videos into spatial audio is especially challenging due to the large number of mixed sources (voices and instruments). We also identified 489 videos that were recorded with a "horizontal" spatial audio microphone (i.e. only containing $\phi_w(t)$, $\phi_x(t)$ and $\phi_y(t)$ channels). In this case, we simply ignore the Z channel $\phi_z(t)$ when computing each metric including the STFT loss. Fig. 2 shows illustrative video frames and summarizes the most common categories for each dataset.

## 4   Evaluation

For our experiments, we randomly sample three partitions, each containing 75% of all videos for training and 25% for testing. Networks are trained to generate audio at 48kHz from input mono audio processed at 48kHz and video at 10Hz. Each training sample consists of a chunk of 0.6s of mono audio and a single frame of RGB and flow, which are used to predict 0.1s of spatial audio at the center of the 0.6s input window. To make the model more robust and remove any bias to content in the center, we augment datasets during training by randomly rotating both video and spatial audio around the vertical (z) axis. Spatial audio can be rotated by multiplying the ambisonic channels with the appropriate rotation matrix as described in [30], and video frames (in equirectangular format) can be rotated using horizontal translations with wrapping. Networks are trained by back-propagation using the Adam optimizer [28] for 150k iterations (roughly two days) with parameters $\beta_1 = 0.9$, $\beta_2 = 0.999$ and $\epsilon = 1e - 8$, batch size of 32, learning rate of $1e - 4$ and weight decay of 0.0005. During evaluation, we predict a chunk of 0.1s for each second of the test video, and average the results across all chunks. Also, to avoid bias towards longer videos, all evaluation metrics are computed for each video separately, and averaged across videos.

| | REC-STREET | | | YT-CLEAN | | | YT-MUSIC | | | YT-ALL | | |
|---|---|---|---|---|---|---|---|---|---|---|---|---|
| | STFT | ENV | EMD | STFT | ENV | EMD | STFT | ENV | EMD | STFT | ENV | EMD |
| SPATIAL PRIOR | 0.187 | 0.958 | 0.492 | 1.394 | 2.063 | 1.478 | 4.652 | 4.355 | 3.479 | 2.691 | 3.394 | 2.246 |
| U-NET BASELINE | 0.180 | 0.935 | 0.449 | 1.361 | 2.039 | **1.403** | 4.338 | 4.678 | 2.855 | 2.658 | 3.239 | 2.137 |
| OURS-NOVIDEO | 0.178 | 0.973 | 0.450 | 1.370 | 2.081 | 1.428 | 4.220 | 4.591 | 2.654 | 2.635 | 3.200 | 2.117 |
| OURS-NORGB | **0.158** | 0.779 | **0.425** | **1.339** | 1.847 | **1.405** | 3.664 | 3.569 | 2.432 | 2.546 | 2.907 | 2.063 |
| OURS-NOFLOW | 0.172 | 0.784 | 0.440 | **1.349** | 1.778 | **1.402** | 3.615 | 3.467 | 2.403 | 2.455 | 2.665 | **2.023** |
| OURS-NOSEP | **0.152** | 0.790 | **0.422** | 1.381 | **1.773** | 1.415 | 3.627 | 3.602 | 2.447 | **2.435** | 2.694 | 2.050 |
| OURS-FULL | **0.158** | **0.767** | **0.419** | 1.379 | **1.776** | 1.417 | **3.524** | **3.366** | **2.350** | 2.447 | **2.649** | **2.019** |

**Table 1: Quantitative comparisons.** We report three quality metrics (Sec 3.3): Envelope distance (ENV), Log-spectral distance (LSD), and earth-mover's distance (EMD), on test videos from different datasets (Sec 3.4). Lower is better. All results within 0.01 of the top performer are shown in bold.

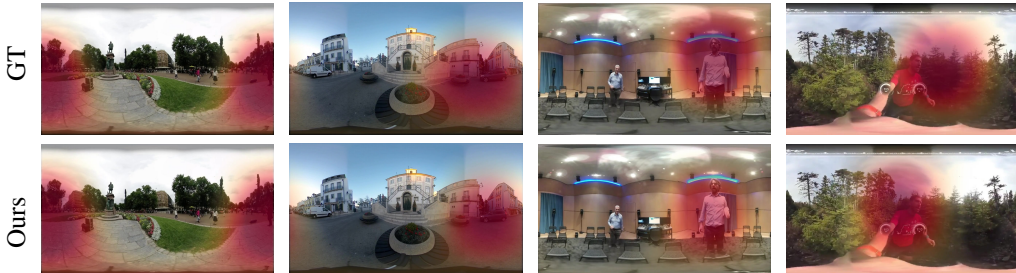

**Figure 3: Qualitative Results.** Comparison between predicted and recorded FOA. Spatial audio is visualized as a color overlay over the frame, with darker red indicating locations with higher audio energy.

**Real time performance** The proposed procedure can generate 1s of spatial audio at 48000Hz sampling rate in 103ms, using a single 12GB Titan Xp GPU (3840 cores running at 1.6GHz).

**Baselines** Since spatial audio generation is a novel task, no established methods exist for comparison purposes. Instead, we ablate our architecture to determine the relevance of each component, and compare it to the prior spatial distribution of audio content and a popular, domain-independent baseline architecture. Quantitative results are shown in Table 1.

To determine the role of the visual input, we remove the RGB encoder (NORGB), the flow encoder (NOFLOW), or both (NOVIDEO). We also remove the separation block entirely (NOSEP), and multiply the localization weights with the input mono directly. The results indicate that the network is highly relying on visual features, with NOVIDEO being one of the worse performers overall. Interestingly, most methods performed well on REC-STREET and YT-CLEAN. However, the visual encoder and separation block are necessary for more complex videos as in YT-MUSIC and YT-ALL.

Since the main sound sources in 360° videos often appear in the center, we validate the need for a complex model by directly using the prior distribution of audio content (SPATIAL-PRIOR). We compute the spatial prior $\bar{E}(\theta)$ by averaging the energy maps $E(\theta, t)$ of (Eq. 5) over all videos in the training set. Then, to induce the same distribution on test videos, we decompose $\bar{E}(\theta)$ into its spherical harmonics coefficients $(c_w, c_x, c_y, c_z)$ and upconvert the input mono using $(\phi_w(t), \phi_x(t), \phi_y(t), \phi_z(t)) = (1, c_x/c_w, c_y/c_w, c_z/c_w) i(t)$. As shown in Table 1, relying solely on the prior distribution is not enough for accurate ambisonic conversion.

We finally compare to a popular encoder-decoder U-NET architecture, which has been sucessfully applied to audio tasks such as audio super-resolution [31]. This network consists of a number of convolutional downsampling layers that progressively reduce the dimension of the signal, distilling higher level features, followed by a number of upsampling layers to restore the signal's resolution. In each upsampling layer, a skip connection is added from the encoding layer of equivalent resolution. To generate spatial audio, we modify the U-NET architecture by setting the number of units in the output layer to the number of ambisonic channels, and concatenate video features to the U-Net bottleneck (i.e., the lowest resolution layer). Our approach significantly outperforms the U-NET architecture, which demonstrates the importance of an architecture tailored to the task of spatial audio generation.

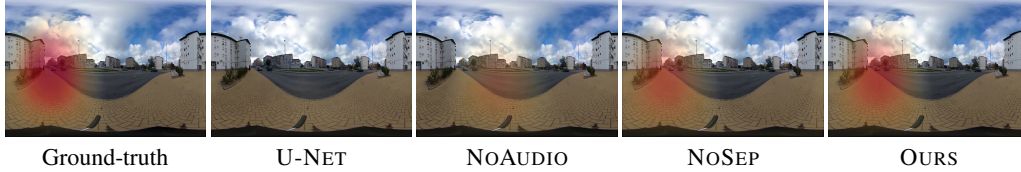

| Ground-truth | U-NET | NOAUDIO | NOSEP | OURS |

Figure 4: **Comparisons.** Predicted FOA produced by different procedures.

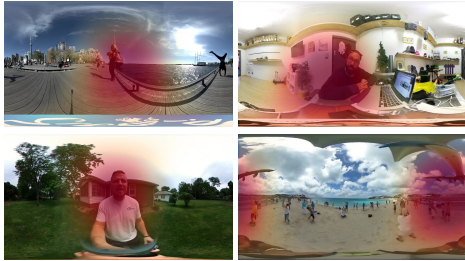

Figure 5: **Mono recordings.** Predicted FOA on videos recorded with a real mono microphone (unknown FOA).

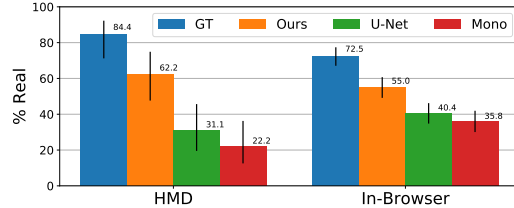

Figure 6: **User studies.** Percentage of videos labeled as "Real" when viewed with audio generated by various methods (GT, OURS, U-NET and MONO) under two viewing experiences (using a HMD device, and in-browser viewing). Error bars represent Wilson score intervals [45] for a 95% confidence level.

**Qualitative results**    Designing robust metrics for comparing spatial audio is an open problem, and we found that only so much can be determined by these metrics alone. For example, fully flat predictions can have a similar EMD to a mis-placed prediction, but perceptually be much worse. Therefore, we also rely on qualitative evaluation and a user study. Fig. 3 shows illustrative examples of the spatial audio output of our network, and Fig. 4 shows a comparison with other baselines. To depict spatial audio, we overlay the directional energy map $E(\boldsymbol{\theta}, t)$ of the predicted ambisonics (Eq. 5) over the video frame at time $t$. As can be seen in most of these examples, our network generates spatial audio that has a similar spatial distribution of energy as the ground truth. Furthermore, due to the form of the audio generator, the sound fidelity of the original mono input is carried over to the synthesized audio. These and other examples, together with the predicted spatial audio, are provided in Supp. material.

The results shown in Table 1 and Fig. 3 use videos recorded with ambisonic microphones and converted to mono audio. To validate whether our method extends to real mono microphones, we scraped additional videos from YouTube that were *not* recorded with ambisonics, and show that we can still generate convincing spatial audio (see Fig. 5 and Supp. material).

**User study**    The real criteria for success is whether viewers believe that the generated audio is correctly spatialized. To evaluate this, we conducted a "real vs fake" user study, where participants were shown a 360° video and asked to decide whether the perceived location of the audio matches the location of its sources in the video (real) or not (fake). Two studies were conducted in different viewing environments: a popular in-browser 360° video viewing platform (YouTube), and with a head-mounted display (HMD) in a controlled environment. We recruited 32 participants from Amazon Mechanical Turk for the in-browser study. For the HMD study, we recruited 9 participants (aged between 20 and 32, 1 female) through an engineering school email list of a large university. In both cases, participants were asked to have normal hearing, and to listen to the audio using headphones. In the HMD study, participants were asked to wear a KAMLE VR Headset. To familiarize participants with the spatial audio experience, each participant was first asked to watch two versions of a pre-selected video with and without correct spatial audio. After the practice round, participants watched 20 randomly selected videos whose audio was generated by one of four methods: GT, the original ground-truth recorded spatial audio; MONO, just the mono track (no spatialization); U-NET, the baseline method; and OURS, the result of our full method. After each video, participants were asked to decide whether its audio was real or fake. In total, 280 clips per method were watched for the in-browser study, and 45 per method in the HMD study.

The results of both studies, shown in Fig 6, support several conclusions. First, our approach outperforms the U-NET baseline and MONO by statistically significant margins in both studies.

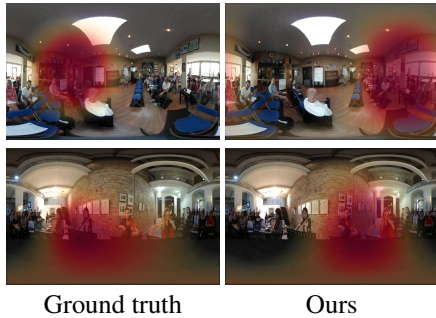

|  | Ground truth | Ours |

| | MONO → FOA | FOA → SOA |
|---|---|---|
| ENV | 1.870 | **0.333** |
| LSD | 3.228 | **0.513** |
| EMD | 1.400 | **0.232** |

**Figure 7: Limitations.** Our algorithm predicts the wrong people who are laughing in a room full of people (top), and the wrong violin who is currently playing in the live performance (right).

**Figure 8: Higher order ambisonics.** (Top) Examples from our synthetic FOA to SOA conversion experiment. (Bottom) Comparison between Mono to FOA and FOA to SOA conversion tasks.

Second, in comparison to in-browser video platforms, HMD devices offer a more realistic viewing experience, which enables non-spatial audio to be identified more easily. Thus, participants were convinced by the ambisonics predicted by our approach at higher rates while wearing an HMD device (62% HMD vs. 55% in-browser). Finally, spatial audio may not always be experienced easily, e.g., when the video does not contain clean sound sources. As a consequence, even videos with GT ambisonics were misclassified in both studies at a significant rate.

## 5 Discussion

**Limitations** We observe several cases where sound sources are not correctly separated or localized. This occurs with challenging examples such as those with many overlapping sources, reverberant environments which are hard to separate, or where there is an ambiguous mapping from visual appearance to sound source (such as multiple, similar looking cars). Fig. 7 shows a few examples. While general purpose spatial audio generation is still an open problem, we provide a first approach. We hope that future advances in audio-visual analysis and audio generation will enable more robust solutions. Also, while total amount of content (in hours) is on par with other video datasets, the number of videos is still low, due to the limited number of 360° video with spatial audio available from online sources. As this number increases, our method should also improve significantly.

**Future work** Although hardware trends change and we begin to see commercial cameras that include spatial audio microphone arrays capable of recording FOA, we believe that up-converting to spatial audio will remain relevant for a number of reasons. Besides the spatialization of legacy recordings with only mono or stereo audio, our method can be used to further increase the ambisonics spatial resolution, for example by up-converting first into second-order ambisonics (SOA). Unfortunately, ground-truth SOA recordings are difficult to collect in-the-wild, since SOA microphones are rare and expensive. Instead, to demonstrate future potential, we applied our approach to the FOA to SOA conversion task, using a small synthetic dataset where pre-recorded sounds are placed at chosen locations, which move over time in random trajectories. These are accompanied by an artificially constructed video consisting of a random background image with identifying icons synchronized with the sound location (see Fig. 8). The results shown in Fig. 8 indicate that converting FOA into SOA may be significantly easier than ZOA to FOA. This is because FOA signals already contain substantial spatial information, and partially separated sounds. Given these findings, a promising area for future work is to synthesize a realistic large scale SOA dataset for learning to convert FOA into high-order ambisonics in order to support more realistic viewing experience.

**Conclusion** We presented the first approach for up-converting conventional mono recordings into spatial audio given a 360° video, and introduced an end-to-end trainable network tailored to this task. We also demonstrate the benefits of each component of our network and show that the proposed generator performs substantially better than a domain independent baseline.

**Acknowledgments** This work was partially funded by graduate fellowship SFRH/BD/109135/2015 from the Portuguese Ministry of Sciences and Education and NRI Grant IIS-1637941.

## Footnotes

*Contact author: `pmaravil@eng.ucsd.edu`

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
