[Supplementary Material · appendix-architecture.pdf]



**Figure 1: Audio encoder**. Detailed representation of audio encoder architecture. Forward pass is left to right.

**Figure 2: Separation**. Detailed representation of source separation architecture. Forward pass is left to right.

**Figure 3: Localization**. Detailed representation of localization architecture. Forward pass is left to right.

# A   Appendix

## A.1   Network Architectures

Both video and flow encoders use the ResNet-18 architecture up to the last convolutional layer. Then, a 1x1 convolutional layer reduces the dimensionality of the feature maps to 128, the resulting maps of size 7x14x128 are flattened, and a fully-connected layer computes a 512-dimensional feature vector. Flow features are extracted from flow maps represented by three channels: the X and Y displacements, as well as the magnitude of the corresponding velocity vector. The audio encoder is a 5 layer CNN applied to the input STFT and detailed in Fig. 1. The audio encoder outputs a 1024-dimensional feature representation for the input audio.

The concatenated audio and video features are then fed to the separation and localization blocks, shown in Figs. 2 and 3, respectively. The separation block (Fig. 2) outputs the $k = 32$ frequency activation maps $\mathbf{a}^{(i)}(\omega, t)$ to be used for modulation of the input STFT $\mathbf{\Phi}(t; \omega)$, and separated wave-forms $f^i(t)$ are computed by inverse STFT, i.e.,

$$f^i(t) = STFT^{-1}\left\{\mathbf{a}^{(i)}(\omega, t) \cdot \mathbf{\Phi}(t; \omega)\right\}$$

In our implementation, the number of frequency components is 1024. The localization block (Fig. 3) outputs, for each of the $k = 32$ sources, the 3 localization weights $\mathbf{w}^i$ associated with the three ambisonics channels $\boldsymbol{\phi} = (\phi_x, \phi_y, \phi_z)$.

Given the localization weights $\mathbf{w}^i(t)$ and separated wave-forms $f^i(t)$, the FOA are generated by

$$\boldsymbol{\phi}(t) = \sum_{i=1}^{k} \mathbf{w}^i(t) f^i(t).$$