[Reviews · NeurIPS 2018]

Reviewer 1



This work proposed a system that converts mono audio into spatial audios while given visual input. The system learns in a self-supervised way from 360 videos. Strength: (1) This paper is the first to solve the problem of spatial audio generation, and adopts the popular audio-visual self-supervision. The formulation and proposed model makes good sense. (2) Both objective and subjective evaluations are performed to verify the effectiveness of the model. (3) Applications of this work are straightforward: converting legacy mono-audio 360 videos into spatial ones for AR/VR applications. Questions: (1) In the proposed architecture, it is not clear where and what is the loss function. As there are two tasks, separation and localization, how are the losses designed for each task? The authors should give details on that. (2) How is the spatial localization done? It seems that the output of RGB and Flow streams have already thrown away the spatial information. (2) How are the ground truth energy maps generated in Figure 3? (3) Notations in the paper is a bit confusing, eg. How are w, f, \Phi_(t, w), phi_(t) in the model (Figure 1) correspond to notations in Equation 1 and 2? Unifying the notations is needed to improve the clarity. (4) More details are needed so that results can be reproduced by other researchers, eg. audio and video frame rate, input audio length, loss functions.

Reviewer 2



The paper proposes an automatic method to spatialise mono-channel audio for 360degree videos, meaning that the output of the method are several audio channels (most commonly two when human-centered applications are targetted) from which one can understand where the sound source is located. The method is based on a CNN which inputs the video and the mono-channel, and that is trained with datasets for which the ambisonic sound tracks are available. The network has a specific path for sound, for RGB and for optical flow. RGB and flow are processed and the fused, further on with audio, in order to obtain an audio-visual embedding of the scene. This embedding is fed to the audio decoder, which produces a set of TF attenuation maps (one per source). The very same audio-visual embedding is used to produce the localization weights, which are combined with the STFT attenuated by the output attenuation maps. This combination allows to generate the spatialized (STFT and hence) wavemors. I enjoyed reading the paper, the application is definitely useful, the methodology interesting, and the text reads well. However, I think the paper needs to be more sharp in some aspects that, right now, may be a bit fuzzy/confusing. For instance, all the claims regarding the novelty of the paper should be very precise. This study is perhaps the first addressing spatialisation from video and mono-audio, but not the first work addressing spatialisation (the audio processing literature is populated with such studies). I do not know how restrictive is to assume that "monaural audio is recorded with an omnidirectional microphone" (L122). I believe that many comercial recording devices do NOT satisfy this constraint, thus clearly limiting the applicative interest of the method. How could this be done for non-omnidirectional microphones? The authors assume that it is required to separate $k$ sources. I think an analysis of what happens when there are more or less sources in the scene is required to understand the impact of the choice of k. Clearly, not all scenarios have the same number of sources. For instance, could (a variation of) the system be trained for k sources plus a "trash" extra source so that it can handle more than k sources? What happens when there are less sources: how do the extra separated sources sound? There is a small notation problem that carries an underlying conceptual problem. The authors should write very clearly that the input of the network is i(t) and not \phi_w(t). I understand that under the right conditions, the former is a very good surrogate of the later, but in any case they are conceptually the same. Therefore I strongly suggest that the manuscript is revised in this direction. When discussing the audio separation, the authors describe the inversion procedure very roughly. This deserve more details (for instance, I understand you use ISTFT and not only IFFT). Do you overlap-add? This details deserve to be mentioned. The very same happens with the envelope distance: its description is to rough not allowing for reproducibility. Finally the reference format should be uniformized.

Reviewer 3



This work proposes a new problem of generating spatial audio for a 360 video with mono audio. To achieve the goal, the author proposes a neural network model that jointly analyzes the audio and video signals and generates the coefficients in the first-order ambisonics format, which is a commonly used spatial audio format. Because this is a new problem, the author collected two 360 video datasets with spatial audio ground truth for both training and evaluation and proposed three evaluation metrics for the problem. The author also perform user study on the generated audio to justify the results. The main contribution of the work is the novel problem. While it is widely believed that spatial audio is important for 360 video viewing, most 360 videos contain only mono audio as described in the paper. This is an important problem in practice, considering the large amount of 360 videos being generated and existing online. This work addresses the problem by generating spatial audio from mono audio, which provides a cheaper solution compared with spatial audio recording and is applicable for existing 360 videos that do not contain spatial audio. Another strength of the work is the evaluation. In particular, the user study clearly justifies not only the superior performance of the model but also the importance of spatial audio. The ablation study also provides further insights into both the problem and the model. On the other hand, I have two concerns about the work. The first is the missing details of the training procedure. In L146, the author mentioned that the model is initialized with weights pre-trained on ImageNet. However, it is unclear how to pre-train the weights for the motion networks on a static image dataset. In L214, the author mentioned that the dataset is randomly split into 4 partition. Is the splitting based on video or on training examples? The second concern is the lack of baseline methods. The author only performs ablation study of the proposed model but does not include simpler baselines. Because the content distribution in 360 video is usually highly biased, it is possible that a simple method (linear regression, etc.) or even the prior distribution on the dataset can achieve a good performance. Although the author addressed this during training (L219), it is unclear whether the superior performance comes from the model or simply the biased distribution in the test set. Simple baselines are still needed to justify that the problem requires a complex learning model. === The author response addresses the concerns I have, except that I think a simpler learning based baseline is still necessary. Although the prior distribution baseline provides much insight for the problem, a simple baseline can help to gauge the performance and make the evaluation more complete.